# Harmonizing Heartbeats: The Mosaic of Cardiac Resynchronization Therapy Responders—A Comprehensive Exploration of Diverse Criteria and Predictors

**DOI:** 10.3390/jcm13164938

**Published:** 2024-08-21

**Authors:** Elke Boxhammer, Sophie Zauner, Johannes Kraus, Christian Dinges, Christiana Schernthaner, Franz Danmayr, Tobias Kolbitsch, Christina Granitz, Lukas J. Motloch, Matthias Hammerer, Michael Lichtenauer, Uta C. Hoppe, Bernhard Strohmer

**Affiliations:** 1Department of Internal Medicine II, Division of Cardiology, Paracelsus Medical University Hospital of Salzburg, 5020 Salzburg, Austria; 2Department of Cardiovascular and Endovascular Surgery, Paracelsus Medical University Hospital of Salzburg, 5020 Salzburg, Austria; c.dinges@salk.at; 3Department of Internal Medicine, Division of Cardiology, Nephrology and Intensive Care Medicine, Salzkammergut Clinic Vöcklabruck, 4840 Vöcklabruck, Austria; l.motloch@salk.at

**Keywords:** cardiac resynchronization therapy, heart failure, left ventricular ejection fraction, proBNP, responder status

## Abstract

**Background:** Heart failure (HF) remains a challenging healthcare issue necessitating innovative therapies like cardiac resynchronization–defibrillation therapy (CRT-D). However, the definition of a CRT-D response lacks uniformity, impeding effective clinical evaluation. This study explores diverse CRT-D responder definitions encompassing functional, echocardiographic and laboratory criteria. **Materials & Methods:** A single-center study involving 132 CRT-D patients scrutinized responder criteria including NYHA stage, LVEF increase and proBNP decrease. Statistical analyses such as Kaplan–Meier curves and Cox hazard regression were employed to evaluate responder characteristics and survival outcomes. **Results:** Responder rates varied across criteria, revealing nuanced patient profiles. CRT-D responders defined by NYHA decrease, LVEF increase or proBNP decrease exhibit improved survival rates after 2 and 3 years (*p* < 0.050). Young age, absence of recent myocardial infarction and normal right ventricular echocardiographic parameters emerge as predictors for positive response. In part, drug-based HF therapy correlates with increased responder rates. Cox regression identified LVEF ≥ 5% and proBNP decrease ≥ 25% as independent predictors of extended survival. **Conclusions:** CRT-D responder definitions exhibit considerable variability, emphasizing the need for a nuanced patient-centered approach. Factors like right ventricular function, drug therapy, atrial fibrillation and renal function influence responses. This study enriches our understanding of CRT-D response and contributes to the foundation for personalized HF management.

## 1. Introduction

Heart failure (HF) stands as a formidable challenge in contemporary healthcare, demanding innovative therapeutic approaches to enhance patient outcomes [1]. Among the myriad interventions, cardiac resynchronization therapy (CRT-D) has emerged as a cornerstone in the management of HF, offering a ray of hope to those afflicted by this complex syndrome [2]. As the field of cardiology continues to evolve, an essential question resonates in the corridors of clinical practice: what defines a responder to CRT-D?

The concept of CRT-D responsiveness has been a subject of extensive investigation. While the benefits of CRT-D are well established, the criteria defining a positive response remain elusive and multifaceted [3,4].

One commonly cited definition of CRT-D responders revolves around objective measures of cardiac function. Traditional metrics such as left ventricular ejection fraction (LVEF) and end-systolic volume have been pivotal in gauging the success of CRT-D [5,6]. Patients experiencing a significant improvement in these parameters post-implantation are often categorized as responders. However, the simplicity of these criteria belies the intricacies of the patient population and the diverse factors influencing response.

Beyond the realms of cardiac mechanics, the clinical realm introduces a myriad of subjective factors that contribute to the definition of CRT-D responders. Symptomatic relief, reflected in improvements in exercise tolerance, reduction in HF hospitalizations and enhancements in quality of life, serves as an invaluable endpoint [3,7,8]. The subjective nature of these outcomes, however, adds a layer of complexity, as individual patient experiences and perceptions come into play, challenging the establishment of uniform criteria.

Furthermore, the emergence of advanced imaging modalities has provided clinicians with an unprecedented glimpse into the myocardial substrate. Tissue Doppler imaging [9], strain imaging [10] and myocardial perfusion imaging [11] are among the techniques that have been explored to refine the identification of CRT-D responders. These modalities offer insights into regional myocardial function and viability, contributing to a more nuanced understanding of response patterns.

As the horizon of CRT-D expands, the role of biomarkers in predicting response has garnered increasing attention. Neurohormonal activation, inflammatory markers and genetic predispositions are among the factors under scrutiny, with ongoing research aiming to elucidate their predictive value [12,13,14]. Integrating these biomarkers into the CRT-D responder definition not only adds a layer of precision but also paves the way for personalized medicine in the realm of HF management.

In this manuscript, we focus specifically on patients with CRT-D rather than those with CRT-P, despite the potential for overlap in the endpoints of interest. This choice stems from the distinct clinical profile and management strategies associated with CRT-D patients who not only receive the benefits of resynchronization but also the added protection against sudden cardiac death due to the defibrillation component. The dual functionality of CRT-D introduces unique considerations in both response criteria and patient outcomes, making it crucial to analyze this cohort independently to derive insights that are directly applicable to this population. By focusing on CRT-D patients, we aim to unravel the intricate tapestry of CRT-D responders, navigating through diverse definitions and shedding light on the complexities that confront clinicians in their pursuit of optimal patient care.

## 2. Material and Methods

### 2.1. Study Population

The study population included 136 patients with indication for an implantation of a CRT-D system at Paracelsus Medical University Hospital, Salzburg, in the period from 2011 to 2021. Four patients were excluded from the final analysis due to loss to follow-up, resulting in a final cohort of 132 patients. The inclusion of patients was consecutive and retrospective. The decision to analyze patients with CRT-D rather than those with CRT-P was based on the distinct clinical characteristics and management strategies associated with CRT-D therapy. CRT-D provides the benefits of cardiac resynchronization along with a defibrillator component, offering protection against sudden cardiac death. This dual functionality introduces unique considerations in response assessment and patient outcomes, making a focused analysis of this population necessary to derive insights directly relevant to their specific clinical outcomes and response criteria.

The study protocol received approval from the local ethics committee of Paracelsus Medical University Salzburg (415-E/2427/7–2019) and adhered to the principles outlined in the Declaration of Helsinki and Good Clinical Practice. Patient consent was waived due to the retrospective nature of the study.

### 2.2. Data Collection

Data were obtained from the ORBIS electronic medical records system (Agfa Healthcare, Version 08043301.04110DACHL) and the medical archiving system (Krankengeschichtsarchiv System, Uniklinikum Salzburg, Softworx by Andreas Schwab TM, 2008) at the University Clinic Salzburg (Austria). The following patient information, including charts and reports from admissions, discharges and laboratory results before and during the CRT-D implantation, was extracted: General clinical data including age, gender and BMI, as well as cardiac risk factors and conditions such as myocardial infarction and the etiology of the patient’s heart failure. The statement “recent” refers to an event (myocardial infarction, stroke, etc.) that occurred no more than 10 years prior to the CRT-D implantation.NYHA stage, which was evaluated by clinicians.Premedication, especially RAAS-blocking agents, betablockers, Ivabradin, MRA, loop diuretics, Amiodaron and SGLT2 inhibitors, as well as Digoxin/Digitoxin.ECG, which was conducted preoperatively.Laboratory values, such as creatinine and proBNP.

### 2.3. Transthoracic Echocardiography

Transthoracic echocardiography (TTE) was routinely conducted, typically 1–4 weeks prior to the CRT-D implantation, utilizing either an iE33 or Epiq 7 ultrasound device (Philips Healthcare, Hamburg, Germany). A minimum of two experienced clinicians with over 4 years of training in echocardiography carried out these examinations. Left ventricular ejection fraction (LVEF) was computed using the biplane Simpson’s method. The maximum tricuspid regurgitation velocity was obtained using continuous wave Doppler over the tricuspid valve. Right atrial pressure and systolic pulmonary artery pressure (sPAP) were calculated following previously established methods [15]. A follow-up echocardiography was performed at intervals of approximately 6 months after the CRT-D implantation. Aside from the TTE, no imaging diagnostics such as cardiac CT or MRI were routinely performed before implantation. 

### 2.4. Decision to CRT-D Implantation

In this study, the criteria for the CRT-D implantation were systematically defined and applied [16]. The selection process involved a comprehensive evaluation of patients based on established clinical, echocardiographic and electrocardiographic parameters. Clinical criteria included HF symptoms despite optimal medical therapy (at least 3 months of up-titrated HF medication) and reduced left ventricular ejection fraction (LVEF ≤ 35%). Additionally, echocardiographic assessments considered measures of ventricular dyssynchrony and structural abnormalities. Electrocardiographic criteria involved QRS duration (QRS width ≥ 130 ms) and morphology (left bundle branch block (LBBB) or non-LBBB/IVCD (intraventricular conduction delay)). The detailed methodology for CRT-D implantation eligibility aimed to provide a robust foundation for patient inclusion, ensuring a standardized and rigorous approach in evaluating the efficacy of CRT-D in the study cohort.

### 2.5. CRT-D Implantation

The implantation procedure involved a transvenous placement of all leads through either the left-sided or right-sided cephalic and/or subclavian veins, with connections made to a previously described biventricular pacemaker [17]. The positioning of the left ventricular lead was aimed at the lateral coronary vein; if this was not accessible, alternative options included the posterolateral coronary vein, a posterior vein or an anterolateral vein. During the implantation period of 10 years, various devices and leads from different manufacturers were implanted depending on current availability and the surgeon’s preference.

### 2.6. Responder Criteria

The current guidelines [16] lack a distinct definition for the determination of responder status. Similarly, the existing literature on this topic lacks a standardized approach, creating challenges in making meaningful comparisons. This work endeavors to integrate functional congestion, echocardiographic criteria and laboratory data in an effort to address this gap. Different LVEF cut-off values (5% vs. 10% improvement) were used according to different literature declarations. Therefore, the following definitions were used in this paper:Functional status:1.NYHA—improvement of ≥I stage 6 months after the CRT-D implantation.Echocardiographic status:2.LVEF—increase of 5% 6 months after the CRT-D implantation.3.LVEF—increase of 10% 6 months after the CRT-D implantation.Laboratory status:4.proBNP—decrease of ≥25% 6 months after the CRT-D implantation.

A patient classified in one responder category can also be classified in another one. This multiple categorization allows for a more comprehensive analysis of the results and takes into account the complexity of individual patient profiles.

### 2.7. Aim of the Study

The relevant aim of this study was to evaluate and compare various criteria for defining responders to CRT-D therapy in patients with heart failure. By applying multiple responder definitions, the study aimed to identify the most reliable and clinically relevant predictors of response, thereby enhancing the understanding of which factors are most indicative of therapeutic success across different criteria.

### 2.8. Statistical Analysis

The sample size for this study was determined through a calculation using G*Power 3.1, specifically for a *t*-test within the means test family, employing an a priori power analysis. The optimal sample size, calculated with an effect size (d) of 0.5, an alpha error of 0.05, a power of 0.95 (1 minus beta error) and an allocation ratio of 1, was found to be 176 patients. The current study, with a sample size of 132 patients, achieves a satisfactory power of 0.885 based on the parameters mentioned above.

Statistical analysis and graphical representation were conducted using SPSS (Version 25.0, SPSS Inc., Chicago, IL USA). To assess the normal distribution of variables, the Kolmogorov–Smirnov–Lilliefors test was employed. Metric data that followed a normal distribution were presented as mean ± standard deviation (SD) and analyzed using an unpaired Student’s *t*-test. For metric data that did not exhibit a normal distribution, the median and interquartile range (IQR) were reported, and the Mann–Whitney U-test was utilized for comparing the two groups, while the Kruskal-Wallis test was employed for comparisons involving more than two groups. Categorical data were represented as frequencies and percentages, and the chi-square test was applied for comparisons.

Kaplan–Meier curves, along with corresponding log-rank tests and the documentation of numbers at risk, were generated to discern potential disparities in 1- to 3-year survival between individuals exhibiting responder and non-responder characteristics.

For the calculation of hazard ratios (HRs) and 95% confidence intervals (CIs) related to 1-, 2- and 3-year mortality, univariate Cox proportional hazard regression models were employed, considering various responder statuses. Subsequently, a multivariable Cox regression analysis was conducted to identify independent predictors of mortality. In this process, responder statuses associated with mortality in the univariate analysis (*p* < 0.050) were included, and a backward variable elimination procedure was implemented.

In order to eliminate potential confounding factors affecting the correlation between various responder statuses and clinical characteristics, a univariate binary logistic regression analysis was conducted. Additionally, a z-transformation was applied to metric data for enhanced comparability. Following this, a multivariate binary logistic regression analysis was undertaken to identify independent factors in predicting diverse responder statuses. To achieve this, covariates linked with a positive responder status in the univariate analysis (*p* < 0.050) were included, and a backward variable elimination process was executed.

## 3. Results

### 3.1. Overall Study Cohort and Baseline Characteristics

A total of 132 patients (75.0% men) were enrolled at Paracelsus Medical University Hospital, Salzburg. An overview of the overall baseline characteristics is provided in Table 1.

The average age of the study population was 65.0 ± 9.5 years. The vast majority of CRT-D patients (84.8%) were implanted for primary prophylactic reasons.

### 3.2. Responder Status and Baseline Characteristics

Table 2 provides an overview of the baseline characteristics in relation to the various responder criteria, whereby one patient can fulfill several responder criteria. Forty-five patients, or 34.1% of the total cohort, did not fulfill any of the responder criteria (non-responder status). In total, 22 patients fulfilled one or two criteria (16.7%), 18 patients (13.6%) fulfilled three and finally 25 patients (18.9%) fulfilled all four criteria of a CRT responder.

Considering the functional status based on the NYHA criterion, 43.9% (58 out of 132) of the patients were identified as responders. Within this responder group, individuals were not only significantly younger (62.0 ± 9.8 years vs. 67.5 ± 8.6 years; *p* = 0.001) but also exhibited a notably lower prevalence of myocardial infarction (24.1% vs. 40.5%; *p* = 0.047) and a reduced incidence of atrial fibrillation (AF) (20.7% vs. 43.2%; *p* = 0.006). Analyzing the laboratory parameters, patients with a positive NYHA responder status demonstrated lower levels of creatinine (1.1 ± 0.5 mg/dL vs. 1.3 ± 0.6 mg/dL; *p* = 0.005) and proBNP values (1179.5 ± 2347.3 ng/L vs. 2612.5 ± 3469.8 ng/L; *p* < 0.001).

A comparable trend was observed among patients with a positive responder status for an increase in LVEF of ≥5% and ≥10%. In these instances, individuals were not only younger (LVEF ≥ 5%: 62.1 ± 9.7 years vs. 67.5 ± 8.6 years; *p* = 0.001—LVEF ≥ 10%: 61.4 ± 10.0 years vs. 66.8 ± 8.8 years; *p* = 0.002) but also exhibited a significantly higher body mass index (BMI) (28.9 ± 5.4 kg/m^2^ vs. 26.5 ± 4.3 kg/m^2^; *p* = 0.005—LVEF ≥ 10%: 29.2 ± 5.7 kg/m^2^ vs. 26.8 ± 4.4 kg/m^2^; *p* = 0.020). Similarly, patients with an LVEF elevation of ≥5% and ≥10% had significantly lower incidences of recent myocardial infarction (23.3% vs. 41.7%; *p* = 0.026—LVEF ≥ 10%: 20.9% vs. 39.3%; *p* = 0.036) and atrial fibrillation (18.3% vs. 45.8%; *p* = 0.001—LVEF ≥ 10%: 11.6% vs. 43.8%; *p* < 0.001). Additionally, they exhibited lower levels of creatinine (1.0 ± 0.3 mg/dL vs. 1.4 ± 0.6 mg/dL; *p* < 0.001—LVEF ≥ 10%: 1.0 ± 0.3 mg/dL vs. 1.3 ± 0.6 mg/dL; *p* < 0.001) and proBNP values (1179.5 ± 2222.3 ng/L vs. 2747.5 ± 3833.8 ng/L; *p* < 0.001—LVEF ≥ 10%: 1215.0 ± 2398.0 ng/L vs. 2041.0 ± 3536.5 ng/L; *p* = 0.004).

The identification of a responder status through proBNP revealed a profile of patients who were not only younger (61.6 ± 10.1 years vs. 67.8 ± 8.0 years; *p* < 0.001) but also exhibited a higher BMI (28.8 ± 5.5 kg/m^2^ vs. 26.6 ± 4.3 kg/m^2^; *p* = 0.011) and a lower incidence of AF (18.6% vs. 45.2%; *p* = 0.001). Moreover, as anticipated, this responder group demonstrated a more optimized and comprehensive HF drug therapy. The presence of loop diuretics (57.6% vs. 83.6%; *p* = 0.001) and amiodarone (18.6% vs. 41.1%; *p* = 0.006) was significantly more prevalent in association with a non-responder status.

### 3.3. Responder Status and Follow-Up Characteristics

Table 3 provides a concise overview of the pertinent clinical characteristics observed during the 6-month follow-up.

Irrespective of the responder criteria employed, patients identified as responders consistently exhibited superior control over NYHA progression, creatinine and proBNP values, along with improved left ventricular ejection fraction (LVEF) recorded postoperatively after 6 months. Specifically, patients with a positive responder status for NYHA ≥ I and LVEF ≥ 5% demonstrated notably enhanced right ventricular function, as evidenced by the determination of tricuspid annular plane systolic excursion (TAPSE) (NYHA ≥ I: 19.6 ± 3.1 mm vs. 16.6 ± 4.2 mm; *p* = 0.027—LVEF ≥ 5%: 19.7 ± 3.5 mm vs. 16.1 ± 3.8 mm; *p* = 0.007) or the TAPSE/sPAP ratio, reflecting improved right ventricular–arterial coupling (NYHA ≥ I: 0.6 ± 0.1 vs. 0.4 ± 0.1; *p* < 0.001—LVEF ≥ 5%: 0.5 ± 0.2 vs. 0.4 ± 0.2; *p* = 0.041). Interestingly, neither defibrillator shock therapies nor ventricular tachycardias up to three years had an impact on the investigated responder status. 

### 3.4. Responder Status-Dependent Survival after CRT-D Implantation

To visualize the survival of responders vs. non-responders using the definitions above, Kaplan–Meier curves were generated up to 3 years after CRT-D implantation with corresponding log-rank tests and numbers at risk calculated annually (Figure 1). Patients with a positive responder status, regardless of the chosen definition, exhibited markedly enhanced survival rates in the calculated log-rank tests for 2- and 3-year survival (Figure 1A: NYHA ≥ I; Figure 1B: LVEF ≥ 5%; Figure 1C: LVEF ≥ 10%; Figure 1D: proBNP decrease ≥ 25%). Notably, the 1-year log-rank tests for the responder criterion NYHA ≥ I and LVEF ≥ 5% also demonstrated statistically significant differences.

Cox hazard regression analysis was performed for 1, 2 and 3 years to ascertain the predictive capacity of individual responder criteria or combinations thereof in determining the survival of the recipients of CRT-D (Table 4). 

Concerning 3-year survival, the responder criteria LVEF ≥ 5% (*p* = 0.033) and proBNP decrease ≥ 25% (*p* = 0.041) emerged as independent factors associated with extended survival following the CRT-D implantation.

### 3.5. Predictive Factors Regarding Responder Status

To ascertain a significant statistical association between various responder criteria and other clinical factors, particularly gender, age, weight, height, etc., both univariate and multivariable binary logistic regressions were conducted (Table 5, Table 6, Table 7 and Table 8).

For the functional status criterion of NYHA ≥ I (Table 5), young age (HR: 0.553, 95% CI: 0.306–0.997; *p* = 0.049), the absence of recent MI (HR: 0.217, 95% CI: 0.063–0.743; *p* = 0.015) and preoperative TAPSE (HR: 1.832, 95% CI: 1.014–3.311; *p* = 0.045) were independent factors for a positive response rate.

For the echocardiographic status with an increase in LVEF ≥ 5% (Table 6), the use of an SGLT2 inhibitor (HR: 9.013, 95% CI: 1.614–50.313; *p* = 0.012), a low baseline creatinine (HR: 0.155, 95% CI: 0.047–0.505; *p* = 0.002) and, again, the TAPSE (HR: 2.858, 95% CI: 1.305–6.259; *p* = 0.009) were independent criteria for a positive response after CRT-D treatment. With an improvement in LVEF ≥ 10% (Table 7), the absence of a previous myocardial infarction (HR: 0.091, 95% CI: 0.012–0.667; *p* = 0.018) and the preoperative absence of atrial fibrillation (HR: 0.028, 95% CI: 0.002–0.314; *p* = 0.004) were favorable, independent factors for a positive responder status.

Independent factors associated with the laboratory definition of CRT-D response (Table 8) were increased BMI (HR: 1.545, 95% CI: 1.023–2.332; *p* = 0.039), the absence of AF (HR: 0.369, 95% CI: 0.149–0.918; *p* = 0.032), the use of angiotensin-receptor–neprilysin inhibitor (ARNI) (HR: 2.717, 95% CI: 1.110–6.649; *p* = 0.029) and low baseline creatinine (HR: 0.455, 95% CI: 0.248–0.834; *p* = 0.011).

## 4. Discussion

CRT-D responder definitions exhibit considerable variability, lacking standardization across the medical community. This lack of consensus poses a significant challenge in clinical practice, as diverse criteria are employed to identify responders. The absence of clear, universally accepted responder definitions in the current guidelines [16] further compounds this issue, leaving clinicians without a standardized framework for patient evaluation and CRT-D response assessment. Consequently, the inconsistency in defining responders impedes the comparison of study findings, complicates the establishment of evidence-based practices and hinders effective communication among healthcare professionals. 

The aim of this single center study was to compare different definitions of CRT-D responder status using a wide range of functional, echocardiographic and laboratory criteria. Once more, it becomes evident that there is no singular definition for a CRT-D responder. Instead, numerous parameters must be integrated to allow accurate predictions regarding whether a patient will derive benefits from a CRT-D system.

### 4.1. Differential Impact of LVEF and proBNP on CRT-D Outcomes: Understanding the Discrepancy in Predictive Significance

The findings from our Cox regression analysis reveal some intriguing and seemingly contradictory patterns regarding LVEF improvement and proBNP reduction as responder criteria. Specifically, we observed that an LVEF improvement of greater than 5% was a significant predictor of improved 3-year survival (*p* = 0.033), while a greater than 10% improvement was not significant. Additionally, despite LVEF improvement and proBNP reduction being independent responder criteria, their interaction did not yield significant results.

Significance of LVEF Improvement > 5% vs. > 10%

The observation that an LVEF improvement greater than 5% is statistically significant while a greater than 10% improvement is not might initially seem counterintuitive. However, this can be understood in the context of the clinical characteristics and distribution of the patient population. An improvement of 5% in LVEF captures a broader range of patients (60/132), including those with more moderate heart failure, where even a modest improvement in LVEF translates to significant clinical benefits. In contrast, a 10% improvement is less common and may only occur in patients with more substantial or rapid recovery (43/132), which is a smaller subgroup, leading to less statistical power to detect significance.

Interaction Between LVEF Improvement and proBNP Reduction

Regarding the interaction between LVEF improvement and proBNP reduction, the lack of significance suggests that these two variables may be influencing survival outcomes through partially independent mechanisms, rather than synergistically. LVEF improvement reflects a better mechanical function of the heart, whereas proBNP reduction indicates a decrease in the neurohormonal stress response and fluid overload, both of which are beneficial but may not always occur simultaneously or be directly correlated in the same patients.

In other words, a patient might experience significant clinical improvement with a reduction in proBNP levels due to better fluid management, even if their LVEF does not improve as much. Conversely, an improvement in LVEF could occur without a significant reduction in proBNP, especially if the latter is influenced by other comorbidities such as renal dysfunction. This independence could explain why the interaction between these variables did not reach statistical significance, as their combined effect may not be additive in predicting survival.

Implications for Clinical Practice

These findings highlight the complexity of heart failure management and the challenges in identifying universal markers of response to CRT-D. While both LVEF improvement and proBNP reduction are valuable indicators of response, their independent effects and lack of significant interaction suggest that they should be considered complementary rather than overlapping measures. Clinicians should be aware that patients may benefit from CRT-D in different ways—some through improved cardiac function (LVEF), others through better neurohormonal regulation (proBNP), and that focusing on just one measure may not fully capture the therapeutic benefit in all patients.

### 4.2. Influence of Right Ventricular Function on CRT-D Implantation

The role of right ventricular function in determining CRT-D responder status is a critical aspect deserving thorough discussion. Our findings underscore the significance of assessing right ventricular function, particularly in patients categorized as CRT-D responders based on left ventricular criteria. The right ventricle’s intricate interplay with the left ventricle and its response to CRT-D can significantly influence overall cardiac performance [18].

Several studies have highlighted the impact of right ventricular dysfunction on clinical outcomes in CRT-D recipients [18,19,20,21]. Previous research on ventricular leads and leadless pacemakers has highlighted their impact on right ventricular function, underscoring the need to consider the right ventricle in cardiac interventions. For example, La Fazia et al. [22] demonstrated a low prevalence of new-onset severe tricuspid regurgitation following leadless pacemaker implantation, offering valuable insights that could be relevant to CRT systems. Additionally, research by Sharma et al. [23] has shed light on the dynamics of right ventricular function during cardiac resynchronization therapy (CRT).

In our investigation, the positive association between CRT-D responders, defined by an improvement of functional status (NYHA improvement ≥ I) or by echocardiographic status (LVEF increase ≥ 5%), and preserved right ventricular function preoperatively, as evidenced by TAPSE and the TAPSE/sPAP ratio, further emphasizes the importance of considering both ventricles in evaluating CRT-D efficacy. These results were almost congruent with previous studies by Abreu et al. [24] (TAPSE) and Stassen et al. [25] (TAPSE/sPAP), which also propagated a better response to CRT-D therapy with preserved right ventricular function.

The observed context between a positive CRT-D responder status and a normal right ventricular function prompts a deeper exploration of the potential mechanisms involved. It raises questions about the hemodynamic and electrical interactions between the ventricles and how optimizing CRT-D settings for both may contribute to better overall outcomes. Additionally, these findings advocate for a comprehensive evaluation of both ventricles in CRT-D assessment protocols and underscore the need for future research to elucidate the nuanced interplay between left and right ventricular function in CRT-D responders.

### 4.3. Influence of Drug-Based HF Therapy on CRT-D Implantation

Optimal pharmacological management is integral to the comprehensive care of HF patients, and its influence on the outcomes with CRT-D is a topic of substantial importance [26].

Our study reveals compelling associations between specific drug therapies and CRT-D responder status. Notably, patients on more extensive HF drug regimens demonstrated higher rates of positive CRT-D response. This finding underscores the synergistic relationship between pharmacological interventions and CRT-D efficacy. It suggests that an optimized drug-based HF therapy may create a more favorable substrate for the success of CRT-D, potentially enhancing its clinical benefits [27].

The observed positive context between CRT-D response and certain drug classes, such as beta-blockers, ACEIs, MRAs and ARNIs, aligns with established evidence supporting the efficacy of these medications in HF management. Their impact on neurohormonal modulation and ventricular remodeling likely contributes to the observed association with improved CRT-D outcomes [27,28].

However, the complexities surrounding drug-based HF therapy and CRT-D response warrant careful consideration. The heterogeneity of HF etiologies and patient characteristics introduces variability in drug responses and, consequently, CRT-D outcomes. Furthermore, the intricate interplay between pharmacological and device-based therapies necessitates a personalized and nuanced approach to patient management. In further studies, special attention should be given to documenting medication changes after implantation to draw more precise conclusions on their influence on therapy response. Future research should delve into the specific mechanisms through which individual drug classes influence CRT-D response, exploring potential synergies and interactions. Additionally, investigations into the optimal timing and sequencing of drug therapy initiation in relation to CRT-D implantation could provide valuable insights for refining treatment strategies.

### 4.4. Influence of Atrial Fibrillation on CRT-D Implantation

Atrial fibrillation, as a prevalent comorbidity in HF patients, adds a layer of complexity to the evaluation of CRT-D outcomes. Our study meticulously examined the impact of AF on CRT-D response, recognizing the challenges posed by this arrhythmia in achieving optimal cardiac resynchronization. The observed difference in CRT-D responder rates between patients with and without AF highlights a potential correlation between atrial fibrillation and a less favorable response to CRT-D [29] or increased mortality [30].

One plausible explanation for the reduced responder rates in the AF subgroup lies in the irregular atrial rhythm characteristic of AF. This irregularity can disrupt the temporal relationship between atrial and ventricular contractions, complicating the achievement of optimal biventricular synchronization—a cornerstone of successful CRT-D. The irregular ventricular activation and the loss of atrioventricular synchrony in the presence of AF may contribute to suboptimal CRT-D response [31,32].

The implications extend beyond responder rates to the intricacies of device programming and optimization for patients with atrial fibrillation. Tailoring CRT-D strategies to address the unique challenges posed by irregular atrial rhythm becomes paramount. CRT-D optimizing device settings and adjusting pacing algorithms, together with pharmacological or ablative measures to block AV nodal conduction, may be strategies to improve the outcomes of CRT-D recipients with concomitant AF.

### 4.5. Influence of Kidney Function on CRT-D Implantation

Renal function, as reflected by serum creatinine levels, emerges as a key determinant with potential implications for the outcomes of CRT-D [33]. A notable finding from our study is the inverse correlation between serum creatinine levels and the response to CRT-D, where responders exhibited lower creatinine values. This association has also been reported by Goldenberg et al. [34] as part of the MADIT-CRT-D trial.

Interestingly, while creatinine levels were associated with responder rates, chronic kidney disease (CKD) status was not. This apparent discrepancy warrants further discussion. Creatinine levels provide a direct and continuous measure of kidney function, and subtle elevations in creatinine may reflect early or mild renal impairment that can influence cardiovascular outcomes, even before CKD is clinically apparent. In contrast, CKD is a broader and more categorical diagnosis that may encompass a wide range of kidney function levels, potentially diluting its association with CRT-D response.

The influence of creatinine on CRT-D outcomes might be explained by its role as an indicator of overall metabolic health and its impact on fluid balance, electrolyte homeostasis and neurohormonal activation—factors intricately linked to heart failure progression. Impaired renal function can lead to fluid overload and heightened neurohormonal activity, exacerbating heart failure and potentially diminishing the effectiveness of CRT-D. Additionally, the use of contrast media following CRT implantation, leading to contrast-induced nephropathy, may also influence left ventricular ejection fraction (LVEF) recovery, as demonstrated by Strisciuglio et al. [35].

The observed correlation between lower creatinine levels and better CRT-D outcomes suggests that optimizing renal function may enhance response rates [36]. Strategies aimed at mitigating renal impairment, such as meticulous fluid management and the judicious use of medications, could be integral to improving CRT-D outcomes [37]. Furthermore, CRT-D devices that provide information about lung fluid status through transthoracic impedance measurements may help reduce heart failure hospitalization risks [38].

However, the lack of association between CKD status and CRT-D response introduces complexity. CKD, as a more heterogeneous and less-sensitive marker, may not capture the nuanced variations in renal function that creatinine levels do. This highlights the need for more refined biomarkers or a more detailed assessment of renal function beyond CKD status. Future research should explore advanced renal biomarkers, assess fluid status more precisely and investigate interventions aimed at renal optimization. Understanding the mechanisms through which renal function influences CRT-D response remains a critical area for further exploration.

### 4.6. Summary of Key Findings and Clinical Impact

In summary, the study identifies several key findings with notable clinical implications:Right Ventricular Function: The positive association between preserved right ventricular function and improved CRT-D response highlights the importance of evaluating both ventricles. Clinicians should integrate right ventricular assessments into CRT-D evaluation to better predict patient outcomes and tailor interventions.Pharmacological Therapy: The study reveals that optimized heart failure drug regimens are associated with higher CRT-D response rates. This underscores the need for a coordinated approach that combines effective pharmacological management with CRT-D therapy, potentially enhancing overall treatment efficacy.Atrial Fibrillation: The reduced CRT-D response observed in patients with atrial fibrillation suggests that tailored CRT-D strategies are necessary for this subgroup. Adjustments in device programming or additional treatments may improve outcomes for patients with AF.Renal Function: The inverse correlation between serum creatinine levels and CRT-D response highlights the impact of renal function on therapy outcomes. Optimizing renal function through careful management could play a crucial role in enhancing CRT-D effectiveness.

These findings collectively emphasize the need for a multifaceted approach to CRT-D therapy, incorporating comprehensive assessments and personalized treatment strategies to optimize patient outcomes.

## 5. Limitation

Single-Center Design: The study’s reliance on data from a single center may limit the generalizability of the findings. Variations in patient demographics, local practices and healthcare infrastructure could influence the external validity of the results.Retrospective Nature: The retrospective nature of the study design might introduce inherent biases, including selection bias and information bias. The reliance on existing medical records could lead to incomplete or missing data, impacting the comprehensiveness of the analysis.Sample Size: The study’s sample size, though sufficient for the conducted analyses, might pose limitations when stratifying results based on certain subgroups or rare outcomes. Larger cohorts would enhance the statistical power for subgroup analyses, even if the statistical power was a satisfactory 88.5%.Definition of Responder Status: The lack of a universally accepted definition for CRT-D responder status could introduce variability in patient classification. The absence of standardized criteria across studies or clinical guidelines may impact the consistency and comparability of findings.Follow-Up Duration: The study’s follow-up duration may be limited, particularly if exploring longer-term outcomes. Extended follow-up periods could provide a more comprehensive understanding of the durability of CRT-D response and potential late effects.Incomplete Covariate Adjustment: Despite efforts to control for confounding variables, unmeasured or residual confounding may persist. Incomplete adjustment for relevant covariates could influence the accuracy of the observed associations.Medication changes: The impact of CRT-D on medication, including potential post-implantation adjustments, is hindered by the probable unavailability of data on medication changes, with this study solely relying on baseline medication documentation.

## 6. Conclusions

This manuscript embarks on a comprehensive journey, weaving through the intricate fabric of CRT-D responsiveness definitions, acknowledging the challenges faced by clinicians and highlighting the imperative for a holistic and patient-centered approach. As we delve into the nuances of CRT-D response, the hope is to foster a dialogue that transcends conventional boundaries, propelling us toward a future where personalized medicine in HF management becomes a tangible reality.

## Figures and Tables

**Figure 1 jcm-13-04938-f001:**
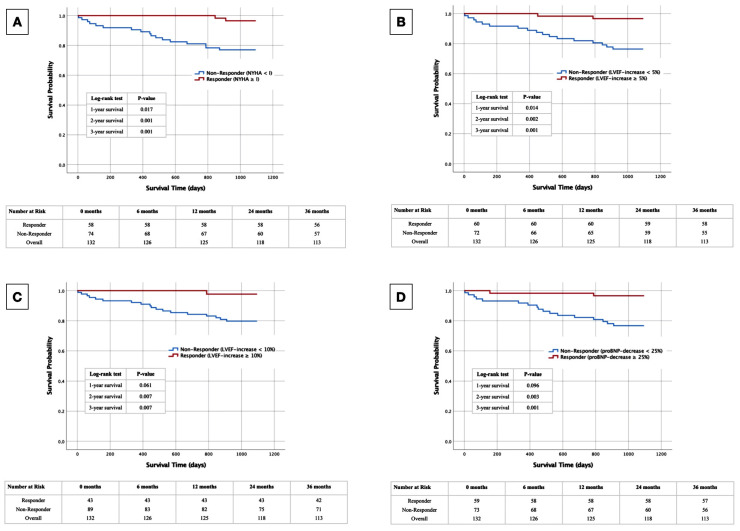
Kaplan–Meier curves with corresponding numbers at risk and annual log-rank tests for detection of 1- to 3-year survival in CRT-D responders vs. CRT-D non-responders. (**A**): Responder criterion NYHA improvement ≥ I. (**B**): Responder criterion LVEF increase ≥ 5%. (**C**): Responder criterion LVEF increase ≥ 10%. (**D**): Responder criterion proBNP decrease ≥ 25%.

**Table 1 jcm-13-04938-t001:** Baseline characteristics of overall study cohort.

Demographics	Overall
*n*	132
Male (%)	75.0
Age (years—mean ± SD)	65.0 ± 9.5
Clinical	
Weight (kg—mean ± SD)	83.5 ± 16.9
Height (m—mean ± SD)	173.7 ± 8.5
BMI (kg/m^2^—mean ± SD)	27.6 ± 5.0
BMI < 18.5 kg/m^2^ (%)	2.3
BMI 18.5–24.9 kg/m^2^ (%)	31.8
BMI 25.0–29.9 kg/m^2^ (%)	38.6
BMI 30.0–34.9 kg/m^2^ (%)	18.9
BMI 35.0–39.9 kg/m^2^ (%)	6.8
BMI ≥ 40.0 kg/m^2^ (%)	1.5
ICMP (%)	35.6
NICMP (%)	59.1
Arterial Hypertension (%)	65.2
Diabetes mellitus (%)	39.4
Dyslipidemia (%)	70.5
CVD (%)	50.8
CVD–1 vessel (%)	21.1
CVD—2 vessels (%)	11.4
CVD—3 vessels (%)	16.7
Recent MI (%)	33.3
Recent CABG (%)	11.4
AF (%)	33.3
COPD (%)	12.9
Asthma (%)	2.3
PAOD (%)	8.3
Anemia (%)	3.8
CKD > II (%)	44.7
Recent Stroke (%)	11.4
Functional Class	
NYHA (median ± IQR)	3.0 ± 1.0
NYHA II (%)	43.9
NYHA III (%)	53.8
NYHA IV (%)	2.3
Medication	
ACEI/ARB (%)	67.4
BB (%)	95.5
Ivabradine (%)	6.8
MRA (%)	72.0
ARNI (%)	28.8
SGLT2I (%)	12.1
Loop Diuretics (%)	72.0
Digoxin/Digitoxin (%)	12.1
Amiodarone (%)	31.1
Laboratory	
Creatinine (mg/dL—median ± IQR)	1.2 ± 0.5
proBNP (ng/L—median ± IQR)	2459.0 ± 3146.5
ECG	
LBBB (%)	88.6
QRS width (ms—mean ± SD)	170.4 ± 28.4
Echocardiography	
LVEF (%—mean ± SD)	27.0 ± 7.6
LVEDD (mm—mean ± SD)	63.9 ± 8.2
TAPSE (mm—mean ± SD	18.4 ± 4.8
sPAP (mmHg—mean ± SD)	45.8 ± 13.2
Implantation characteristics	
Primary prevention (%)	84.8

BMI: body mass index; ICMP: ischemic cardiomyopathy; NICMP: non-ischemic cardiomyopathy; CVD: cardiovascular disease; MI: myocardial infarction; CABG: coronary artery bypass graft; AF: atrial fibrillation; COPD: chronic obstructive pulmonary disease; PAOD: peripheral artery occlusive disease; CKD: chronic kidney disease; NYHA: New York Heart Association; ACEI/ARB: angiotensin converting enzyme inhibitor/angiotensin-II-receptor blocker; BB: beta blocker; MRA: mineralocorticoid-receptor antagonist; ARNI: angiotensin-receptor–neprilysin inhibitor; SGLT2I: sodium–glucose-transporter-2 inhibitor; IQR: interquartile range; ECG: electrocardiogram; LBBB: left bundle branch block; LVEF: left ventricular ejection fraction; LVEDD: left ventricular end diastolic diameter; TAPSE: tricuspid annular plane systolic excursion; sPAP: systolic pulmonary artery pressure.

**Table 2 jcm-13-04938-t002:** Baseline characteristics depending on different responder definitions.

	Functional Status			Echocardiographic Status						Laboratory Status		
	NYHA Improvement ≥ I			LVEF Increase ≥ 5%			LVEF Increase ≥ 10%			proBNP Decrease ≥ 25%		
	R	NR	*p*	R	NR	*p*	R	NR	*p*	R	NR	*p*
Demographics												
n	58	74		60	72		43	89		59	73	
Male (%)	69.0	79.7	0.156	65.0	83.3	0.015	67.4	78.7	0.163	64.4	83.6	0.012
Age (years—mean ± SD)	62.0 ± 9.8	67.5 ± 8.6	0.001	62.1 ± 9.7	67.5 ± 8.6	0.001	61.4 ± 10.0	66.8 ± 8.8	0.002	61.6 ± 10.1	67.8 ± 8.0	0.000
Clinical												
Weight (kg—mean ± SD)	85.8 ± 16.8	81.6 ± 16.8	0.129	86.2 ± 18.9	81.1 ± 14.7	0.083	87.0 ± 19.1	81.7 ± 15.5	0.095	86.2 ± 18.7	81.2 ± 15.1	0.094
Height (m—mean ± SD)	173.5 ± 7.9	173.9 ± 9.0	0.809	172.3 ± 8.6	174.9 ± 8.2	0.075	172.4 ± 8.6	174.3 ± 8.4	0.227	172.7 ± 8.5	174.6 ± 8.4	0.204
BMI (kg/m^2^—mean ± SD)	28.5 ± 5.1	26.9 ± 4.8	0.074	28.9 ± 5.4	26.5 ± 4.3	0.005	29.2 ± 5.7	26.8 ± 4.4	0.020	28.8 ± 5.5	26.6 ± 4.3	0.011
BMI < 18.5 kg/m^2^ (%)	3.4	1.4	0.673	3.3	1.4	0.877	4.7	1.1	0.489	3.4	1.4	0.806
BMI 18.5–24.9 kg/m^2^ (%)	27.6	35.1	0.125	23.3	38.9	0.008	20.9	37.1	0.013	27.1	35.6	0.047
BMI 25.0–29.9 kg/m^2^ (%)	34.5	41.8	0.386	35.0	41.7	0.433	34.8	40.4	0.538	32.2	43.8	0.172
BMI 30.0–34.9 kg/m^2^ (%)	25.9	13.5	0.072	25.0	13.9	0.105	23.3	16.9	0.379	23.7	15.1	0.207
BMI 35.0–39.9 kg/m^2^ (%)	6.9	6.8	0.975	11.7	2.8	0.044	14.0	3.4	0.024	10.2	4.1	0.170
BMI ≥ 40.0 kg/m^2^ (%)	1.7	1.4	0.862	1.7	1.4	0.896	2.3	1.1	0.596	3.4	0.0	0.113
ICMP (%)	32.8	37.8	0.545	30.0	40.3	0.219	30.2	38.2	0.370	28.8	41.1	0.143
NICMP (%)	62.1	56.8	0.538	65.0	54.2	0.207	69.8	53.9	0.083	64.4	54.8	0.264
Arterial Hypertension (%)	62.1	67.6	0.511	61.7	68.1	0.443	60.5	67.4	0.432	71.2	60.3	0.191
Diabetes mellitus (%)	34.5	43.2	0.307	36.7	41.7	0.558	32.6	42.7	0.264	42.4	37.0	0.529
Dyslipidemia (%)	70.7	70.3	0.958	70.0	70.8	0.917	62.8	74.2	0.180	69.5	71.2	0.827
CVD (%)	46.6	52.7	0.392	40.0	59.6	0.024	37.2	57.3	0.030	44.1	56.2	0.167
CVD—1 vessel (%)	25.9	16.2	0.199	23.3	18.1	0.505	23.3	19.1	0.623	27.1	15.1	0.104
CVD—2 vessels (%)	6.9	17.6	0.440	6.7	18.1	0.376	7.0	14.6	0.358	5.1	17.8	0.062
CVD—3 vessels (%)	13.8	18.9	0.393	10.0	22.2	0.051	9.3	20.2	0.103	11.9	20.5	0.161
Recent MI (%)	24.1	40.5	0.047	23.3	41.7	0.026	20.9	39.3	0.036	27.1	38.4	0.173
Recent CABG (%)	8.6	13.5	0.379	8.3	13.9	0.317	11.6	11.2	0.947	8.5	13.7	0.347
AF (%)	20.7	43.2	0.006	18.3	45.8	0.001	11.6	43.8	0.000	18.6	45.2	0.001
COPD (%)	8.6	16.2	0.196	8.3	16.7	0.155	11.6	13.5	0.766	10.2	15.1	0.403
Asthma (%)	5.2	0.0	0.048	3.3	1.4	0.455	4.7	1.1	0.202	3.4	1.4	0.439
PAOD (%)	5.2	10.8	0.245	8.3	8.3	1.000	9.3	7.9	0.779	6.8	9.6	0.561
Anemia (%)	0.0	6.8	0.044	3.3	4.2	0.803	2.3	4.5	0.541	3.4	4.1	0.829
CKD > II (%)	37.9	50.0	0.166	30.0	56.9	0.002	23.3	55.1	0.001	37.3	50.7	0.124
Recent Stroke (%)	8.6	13.5	0.379	10.0	12.5	0.823	7.0	13.5	0.270	8.5	13.7	0.347
Functional Class												
NYHA (median ± IQR)	3.0 ± 1.0	3.0 ± 1.0	0.131	3.0 ± 1.0	3.0 ± 1.0	0.775	3.0 ± 1.0	3.0 ± 1.0	0.744	3.0 ± 1.0	3.0 ± 1.0	0.435
NYHA II (%)	37.9	48.6	0.218	45.0	44.4	0.823	41.9	44.9	0.738	40.7	46.6	0.497
NYHA III (%)	56.9	51.4	0.431	53.3	52.8	0.949	55.8	52.9	0.656	55.9	52.0	0.548
NYHA IV (%)	5.2	0.0	0.048	1.7	2.8	0.670	2.3	2.2	0.977	3.4	1.4	0.439
Medication												
ACEI/ARB (%)	62.1	71.6	0.245	66.7	68.1	0.865	67.4	67.4	0.998	57.6	75.3	0.031
BB (%)	96.6	94.6	0.592	95.0	95.8	0.819	95.3	95.5	0.968	96.6	94.5	0.567
Ivabradine (%)	3.4	9.5	0.174	10.0	4.2	0.186	11.6	4.5	0.128	5.1	8.2	0.478
MRA (%)	72.4	71.6	0.920	73.3	70.8	0.750	74.4	70.8	0.663	81.4	64.4	0.031
ARNI (%)	34.5	24.3	0.201	33.3	25.0	0.292	27.9	29.2	0.877	40.7	19.2	0.007
SGLT2I (%)	15.5	9.5	0.290	20.0	5.6	0.011	16.3	10.1	0.309	18.6	6.8	0.039
Loop Diuretics (%)	63.8	78.4	0.064	61.7	80.6	0.016	60.5	77.5	0.041	57.6	83.6	0.001
Digoxin/Digitoxin (%)	6.9	16.2	0.103	11.7	12.5	0.884	7.0	14.6	0.208	8.5	15.1	0.248
Amiodarone (%)	19.0	40.5	0.008	13.3	45.8	0.000	9.3	41.6	0.000	18.6	41.1	0.006
Laboratory												
Creatinine (mg/dL—median ± IQR)	1.1 ± 0.5	1.3 ± 0.6	0.005	1.0 ± 0.3	1.4 ± 0.6	0.000	1.0 ± 0.3	1.3 ± 0.6	0.000	1.0 ± 0.4	1.3 ± 0.5	0.000
proBNP (ng/L—median ± IQR)	1179.5 ± 2347.3	2612.5 ± 3469.8	0.000	1179.5 ± 2222.3	2747.5 ± 3833.8	0.000	1215.0 ± 2398.0	2041.0 ± 3536.5	0.004	1555.0 ± 2742.0	1925.0 ± 3286.0	0.130
ECG												
LBBB (%)	94.8	83.8	0.047	91.7	86.1	0.317	93.0	86.5	0.270	91.5	86.3	0.347
QRS width (ms—mean ± SD)	167.3 ± 24.2	172.09 ± 31.2	0.255	168.1 ± 23.4	172.4 ± 32.0	0.371	169.0 ± 23.6	171.1 ± 30.5	0.667	172.1 ± 30.5	169.0 ± 26.6	0.539
Echocardiography												
LVEF (%—mean ± SD)	26.3 ± 6.4	27.5 ± 8.5	0.335	25.7 ± 7.6	28.0 ± 7.6	0.092	24.9 ± 6.9	27.9 ± 7.8	0.033	26.5 ± 6.6	27.4 ± 8.4	0.497
LVEDD (mm—mean ± SD)	64.9 ± 8.5	63.1 ± 7.9	0.242	63.9 ± 9.0	63.9 ± 7.6	0.983	63.9 ± 9.5	63.9 ± 7.6	0.958	64.1 ± 7.6	63.8 ± 8.7	0.817
TAPSE (mm—mean ± SD	20.0 ± 4.6	17.1 ± 4.6	0.011	20.5 ± 4.0	17.0 ± 4.9	0.002	20.4 ± 3.8	17.7 ± 5.0	0.035	18.9 ± 4.6	17.9 ± 5.0	0.372
sPAP (mmHg—mean ± SD)	42.7 ± 10.0	47.9 ± 14.6	0.108	46.0 ± 11.6	45.7 ± 13.9	0.928	45.8 ± 12.5	45.8 ± 13.5	0.998	42.7 ± 10.3	47.6 ± 14.4	0.128
Implantation characteristics												
Primary prevention (%)	93.1	78.4	0.019	88.3	81.9	0.308	88.4	83.1	0.433	88.1	82.2	0.344

R: responder; NR: non-responder; BMI: body mass index; ICMP: ischemic cardiomyopathy; NICMP: non-ischemic cardiomyopathy; CVD: cardiovascular disease; MI: myocardial infarction; CABG: coronary artery bypass graft; AF: atrial fibrillation; COPD: chronic obstructive pulmonary disease; PAOD: peripheral artery occlusive disease; CKD: chronic kidney disease; ACEI/ARB: angiotensin converting enzyme inhibitor/angiotensin-II-receptor blocker; BB: beta blocker; MRA: mineralocorticoid-receptor antagonist; ARNI: angiotensin-receptor–neprilysin inhibitor; SGLT2I: sodium–glucose-transporter-2 inhibitor; IQR: interquartile range; proBNP: prohormone of brain natriuretic peptide; ECG: electrocardiogram; LBBB: left bundle branch block; LVEF: left ventricular ejection fraction; LVEDD: left ventricular end diastolic diameter; TAPSE: tricuspid annular plane systolic excursion; sPAP: systolic pulmonary artery pressure.

**Table 3 jcm-13-04938-t003:** Follow-up characteristics (6 months) depending on different responder definitions.

	Functional Status			Echocardiographic Status						Laboratory Status		
	NYHA Improvement ≥ I			LVEF Increase ≥ 5%			LVEF Increase ≥ 10%			proBNP Decrease ≥ 25%		
	R	NR	*p*	R	NR	*p*	R	NR	*p*	R	NR	*p*
Functional Class												
NYHA (median ± IQR)	2.0 ± 1.0	2.5 ± 1.0	0.000	2.0 ± 0.5	2.5 ± 1.0	0.000	1.5 ± 1.0	2.5 ± 1.0	0.000	2.0 ± 0.5	2.5 ± 1.0	0.005
Laboratory												
Creatinine (mg/dL—median ± IQR)	1.0 ± 0.4	1.3 ± 0.8	0.005	1.0 ± 0.3	1.4 ± 0.9	0.000	1.0 ± 0.4	1.3 ± 0.7	0.000	1.0 ± 0.3	1.4 ± 0.7	0.000
proBNP (ng/L—median ± IQR)	629.0 ± 1493.0	2270.5 ± 4582.0	0.000	573.0 ± 2058.0	2623.5 ± 4814.0	0.000	573.0 ± 1577.0	2158.5 ± 3947.8	0.000	489.5 ± 965.0	3374.0 ± 4047.0	0.000
ECG												
QRS width (ms—mean ± SD)	153.1 ± 26.0	165.4 ± 29.8	0.014	154.8 ± 29.0	164.4 ± 28.0	0.057	153.8 ± 26.9	163.0 ± 29.3	0.088	154.6 ± 28.6	164.3 ± 28.3	0.054
CRT-D Analysis												
Biventricular Pacing (%)	97.8	97.1	0.658	97.4	97.0	0.710	97.9	97.3	0.643	98.0	97.5	0.739
sVTs (%)	9.3	10.0	0.890	9.9	9.1	0.362	8.5	12.3	0.161	8.4	13.2	0.064
Appropriate Shock (%)	7.5	8.1	0.601	8.9	9.0	0.391	6.7	8.6	0.493	5.6	10.0	0.064
Echocardiography												
LVEF (%—mean ± SD)	35.8 ± 10.2	28.2 ± 8.1	0.001	36.9 ± 8.9	26.5 ± 7.8	0.000	38.6 ± 8.3	27.6 ± 8.3	0.000	34.3 ± 9.5	29.5 ± 9.7	0.043
LVEDD (mm—mean ± SD)	63.4 ± 8.4	61.5 ± 10.7	0.489	62.8 ± 10.0	62.2 ± 9.1	0.811	61.5 ± 10.7	63.0 ± 9.0	0.606	61.1 ± 6.3	64.1 ± 12.1	0.281
TAPSE (mm—mean ± SD)	19.6 ± 3.1	16.6 ± 4.2	0.027	19.7 ± 3.5	16.1 ± 3.8	0.007	19.8 ± 3.6	17.0 ± 4.0	0.065	18.6 ± 3.9	17.2 ± 4.1	0.322
sPAP (mmHg—mean ± SD)	35.4 ± 7.7	44.3 ± 14.2	0.027	36.4 ± 8.9	44.5 ± 14.4	0.054	35.5 ± 9.0	42.9 ± 13.5	0.145	36.6 ± 8.9	44.6 ± 14.4	0.050
TAPSE/sPAP (mean ± SD)	0.6 ± 0.1	0.4 ± 0.1	0.000	0.5 ± 0.2	0.4 ± 0.2	0.041	0.6 ± 0.2	0.4 ± 0.2	0.030	0.5 ± 0.2	0.4 ± 0.2	0.160

R: responder; NR: non-responder; IQR: interquartile range; ECG: electrocardiogram; SD: standard deviation; CRT-D: cardiac resynchronization therapy defibrillator; sVT: sustained ventricular tachycardia; LVEF: left ventricular ejection fraction; LVEDD: left ventricular end diastolic diameter; TAPSE: tricuspid annular plane systolic excursion; sPAP: systolic pulmonary artery pressure.

**Table 4 jcm-13-04938-t004:** Univariate and multivariable cox hazard regression analysis detecting 1-, 2- and 3-year mortality in dependence of different responder definitions.

Cox Regression Analysis	Univariate		Multivariable	
	Hazard Ratio (95% CI)	*p*-Value	Hazard Ratio (95% CI)	*p*-Value
1-year survival				
Responder NYHA ≥ I	54.232 (0.119–24,780.897)	0.201		
Responder LVEF ≥ 5%	57.265 (0.128–25,544.830)	0.128		
Responder LVEF ≥ 10%	38.426 (0.053–27,604.409)	0.277		
Responder proBNP	5.037 (0.606–41.843)	0.134		
Responder NYHA + LVEF ≥ 5%	38.426 (0.053–27,604.409)	0.277		
Responder NYHA + LVEF ≥ 10%	31.401 (0.021–46,570.928)	0.355		
Responder NYHA + proBNP	34.276 (0.034–34,616.859)	0.317		
Responder LVEF ≥ 5% + proBNP	35.565 (0.040–31,656.799)	0.303		
Responder LVEF ≥ 10% + proBNP	30.879 (0.019–50,147.546)	0.363		
2-year survival				
Responder NYHA ≥ I	56.829 (0.758–4260.552)	0.067		
Responder LVEF ≥ 5%	11.831 (1.547–90.457)	0.017	7.044 (0.896–55.342)	0.063
Responder LVEF ≥ 10%	39.429 (0.392–3965.835)	0.118		
Responder proBNP	11.352 (1.485–86.790)	0.019	6.605 (0.841–51.892)	0.073
Responder NYHA + LVEF ≥ 5%	39.429 (0.392–3965.835)	0.118		
Responder NYHA + LVEF ≥ 10%	31.912 (0.193–5275.549)	0.184		
Responder NYHA + proBNP	34.972 (0.276–4437.930)	0.150		
Responder LVEF ≥ 5% + proBNP	36.351 (0.312–4230.689)	0.139		
Responder LVEF ≥ 10% + proBNP	31.358 (0.178–5519.271)	0.192		
3-year survival				
Responder NYHA ≥ I	7.595 (1.754–32.889)	0.007	3.015 (0.622–14.605)	0.170
Responder LVEF ≥ 5%	7.958 (1.838–34.356)	0.006	5.066 (1.135–22.606)	0.033
Responder LVEF ≥ 10%	9.649 (1.288–72.294)	0.027	2.226 (0.135–36.836)	0.576
Responder proBNP	7.651 (1.767–33.124)	0.006	4.768 (1.068–21.278)	0.041
Responder NYHA + LVEF ≥ 5%	40.225 (0.787–2056.868)	0.066		
Responder NYHA + LVEF ≥ 10%	32.311 (0.417–2505.555)	0.117		
Responder NYHA + proBNP	35.520 (0.572–2205.141)	0.090		
Responder LVEF ≥ 5% + proBNP	8.330 (1.112–62.406)	0.039	0.281 (0.009–8.673)	0.468
Responder LVEF ≥ 10% + proBNP	6.036 (0.806–45.223)	0.080		

LVEF: left ventricular ejection fraction.

**Table 5 jcm-13-04938-t005:** Univariate and multivariable binary logistic regression with regard to CRT-D responder criterion NYHA-improvement ≥ I and various clinical characteristics.

CRT-D Responder: NYHA ≥ I Binary Logistic Regression	Univariate		Multivariable	
	Hazard Ratio (95% CI)	*p*-Value	Hazard Ratio (95% CI)	*p*-Value
Gender (male)	0.565 (0.255–1.250)	0.159		
Age	0.536 (0.365–0.788)	0.001	0.553 (0.306–0.997)	0.049
Weight	1.296 (0.910–1.845)	0.151		
Height	0.958 (0.678–1.354)	0.807		
BMI	1.380 (0.966–1.971)	0.077		
ICMP	0.800 (0.389–1.648)	0.546		
NICMP	1.247 (0.618–2.516)	0.538		
Arterial Hypertension	0.785 (0.382–1.613)	0.511		
Diabetes mellitus	0.691 (0.339–1.406)	0.307		
Dyslipidemia	1.020 (0.480–2.168)	0.958		
Cardiovascular Disease (all)	0.740 (0.372–1.475)	0.393		
CVD—1 vessel	1.744 (0.742–4.098)	0.202		
CVD—2 vessels	0.523 (0.171–1.602)	0.257		
CVD—3 vessels	0.663 (0.257–1.710)	0.395		
Recent MI	0.467 (0.218–0.998)	0.049	0.217 (0.063–0.743)	0.015
Recent CABG	0.604 (0.194–1.876)	0.383		
AF	0.342 (0.156–0.750)	0.007	0.611 (0.178–2.091)	0.432
COPD	0.487 (0.161–1.473)	0.203		
Asthma	0.000 (0.000–.)	0.999		
PAOD	0.450 (0.114–1.779)	0.255		
Anemia	0.000 (0.000–.)	0.999		
CKD > II	0.611 (0.304–1.230)	0.167		
Recent Stroke	0.604 (0.194–1.876)	0.383		
NYHA (preoperative)	1.747 (0.909–3.354)	0.094		
ACEI/ARB	0.648 (0.312–1.349)	0.246		
BB	1.600 (0.283–9.056)	0.595		
Ivabradine	0.342 (0.068–1.712)	0.192		
MRA	1.040 (0.483–2.238)	0.920		
ARNI	1.637 (0.767–3.496)	0.203		
SGLT2I	1.758 (0.613–5.045)	0.294		
Loop Diuretics	0.486 (0.225–1.050)	0.066		
Digoxin/Digitoxin	0.383 (0.117–1.257)	0.113		
Amiodarone	0.343 (0.154–0.767)	0.009	1.012 (0.257–3.979)	0.986
Creatinine (baseline)	0.571 (0.366–0.889)	0.013	1.057 (0.457–2.441)	0.897
proBNP (baseline)	0.503 (0.287–0.882)	0.016	0.508 (0.230–1.122)	0.094
LBBB	3.548 (0.951–13.233)	0.059		
QRS width (preoperative)	0.819 (0.576–1.166)	0.268		
LVEF (preoperative)	0.846 (0.596–1.200)	0.349		
LVEDD (preoperative)	1.255 (0.858–1.834)	0.241		
TAPSE (preoperative)	1.951 (1.135–3.355)	0.016	1.832 (1.014–3.311)	0.045
sPAP (preoperative)	0.650 (0.382–1.107)	0.113		
TAPSE/sPAP (preoperative)	1.870 (0.935–3.741)	0.077		
Primary Prevention	3.724 (1.171–11.840)	0.026	2.368 (0.368–15.237)	0.364

CRT-D: cardiac resynchronization therapy; BMI: body mass index; ICMP: ischemic cardiomyopathy; NICMP: non-ischemic cardiomyopathy; CVD: cardiovascular disease; MI: myocardial infarction; CABG: coronary artery bypass graft; AF: atrial fibrillation; COPD: chronic obstructive pulmonary disease; PAOD: peripheral artery occlusive disease; CKD: chronic kidney disease; ACEI/ARB: angiotensin converting enzyme inhibitor/angiotensin-II-receptor blocker; BB: beta blocker; MRA: mineralocorticoid-receptor antagonist; ARNI: angiotensin-receptor–neprilysin inhibitor; SGLT2I: sodium–glucose-transporter-2 inhibitor; proBNP: prohormone of brain natriuretic peptide; LBBB: left bundle branch block; LVEF: left ventricular ejection fraction; LVEDD: left ventricular end diastolic diameter; TAPSE: tricuspid annular plane systolic excursion; sPAP: systolic pulmonary artery pressure.

**Table 6 jcm-13-04938-t006:** Univariate and multivariable binary logistic regression with regard to CRT-D responder criterion LVEF increase ≥ 5% and various clinical characteristics.

CRT-D Responder: LVEF ≥ 5% Binary Logistic Regression	Univariate		Multivariable	
	Hazard Ratio (95% CI)	*p*-Value	Hazard Ratio (95% CI)	*p*-Value
Gender (male)	0.371 (0.164–0.840)	0.017	0.282 (0.041–1.947)	0.199
Age	0.541 (0.368–0.794)	0.002	1.377 (0.627–3.023)	0.425
Weight	1.366 (0.957–1.952)	0.086		
Height	0.725 (0.508–1.036)	0.077		
BMI	1.701 (1.168–2.479)	0.006	1.177 (0.467–2.971)	0.730
ICMP	0.635 (0.308–1.313)	0.221		
NICMP	1.571 (0.777–3.179)	0.209		
Arterial Hypertension	0.755 (0.368–1.549)	0.444		
Diabetes mellitus	0.811 (0.401–1.638)	0.558		
Dyslipidemia	0.961 (0.454–2.035)	0.917		
Cardiovascular Disease (all)	0.450 (0.223–0.904)	0.025	0.358 (0.090–1.418)	0.143
CVD—1 vessel	1.334 (0.571–3.119)	0.505		
CVD—2 vessels	0.488 (0.159–1.493)	0.208		
CVD—3 vessels	0.375 (0.136–1.031)	0.057		
Recent MI	0.426 (0.199–0.911)	0.028	0.480 (0.066–3.483)	0.468
Recent CABG	0.564 (0.181–1.750)	0.321		
AF	0.265 (0.119–0.591)	0.001	0.459 (0.095–2.212)	0.332
COPD	0.455 (0.150–1.373)	0.162		
Asthma	2.448 (0.217–27.682)	0.469		
PAOD	1.000 (0.290–3.454)	1.000		
Anemia	0.793 (0.128–4.909)	0.803		
CKD > II	0.324 (0.157–0.668)	0.002	0.734 (0.110–4.888)	0.749
Recent Stroke	0.778 (0.260–2.325)	0.653		
NYHA (preoperative)	0.899 (0.475–1.703)	0.745		
ACEI/ARB	0.939 (0.452–1.949)	0.865		
BB	0.826 (0.161–4.251)	0.819		
Ivabradine	2.556 (0.611–10.689)	0.199		
MRA	1.132 (0.527–2.434)	0.750		
ARNI	1.500 (0.704–3.197)	0.294		
SGLT2I	4.250 (1.292–13.975)	0.017	9.013 (1.614–50.313)	0.012
Loop Diuretics	0.388 (0.178–0.849)	0.018	0.326 (0.079–1.340)	0.120
Digoxin/Digitoxin	0.925 (0.323–2.650)	0.884		
Amiodarone	0.182 (0.076–0.437)	<0.001	0.395 (0.059–2.645)	0.339
Creatinine (baseline)	0.318 (0.179–0.563)	<0.001	0.155 (0.047–0.505)	0.002
proBNP (baseline)	0.392 (0.206–0.747)	0.004	0.690 (0.140–3.409)	0.649
LBBB	1.774 (0.571–5.510)	0.321		
QRS width (preoperative)	0.855 (0.603–1.213)	0.381		
LVEF (preoperative)	0.737 (0.516–1.053)	0.094		
LVEDD (preoperative)	1.004 (0.693–1.456)	0.983		
TAPSE (preoperative)	2.263 (1.274–4.021)	0.005	2.858 (1.305–6.259)	0.009
sPAP (preoperative)	1.024 (0.618–1.696)	0.926		
TAPSE/sPAP (preoperative)	1.334 (0.654–2.722)	0.428		
Primary Prevention	1.668 (0.619–4.494)	0.311		

CRT-D: cardiac resynchronization therapy; BMI: body mass index; ICMP: ischemic cardiomyopathy; NICMP: non-ischemic cardiomyopathy; CVD: cardiovascular disease; MI: myocardial infarction; CABG: coronary artery bypass graft; AF: atrial fibrillation; COPD: chronic obstructive pulmonary disease; PAOD: peripheral artery occlusive disease; CKD: chronic kidney disease; NYHA: New York Heart Association; ACEI/ARB: angiotensin converting enzyme inhibitor/angiotensin-II-receptor blocker; BB: beta blocker; MRA: mineralocorticoid-receptor antagonist; ARNI: angiotensin-receptor–neprilysin inhibitor; SGLT2I: sodium–glucose-transporter-2 inhibitor; LBBB: left bundle branch block; LVEF: left ventricular ejection fraction; LVEDD: left ventricular end diastolic diameter; TAPSE: tricuspid annular plane systolic excursion; sPAP: systolic pulmonary artery pressure.

**Table 7 jcm-13-04938-t007:** Univariate and multivariable binary logistic regression with regard to CRT-D responder criterion LVEF increase ≥ 10% and various clinical characteristics.

CRT-D Responder: LVEF ≥ 10% Binary Logistic Regression	Univariate		Multivariable	
	Hazard Ratio (95% CI)	*p*-Value	Hazard Ratio (95% CI)	*p*-Value
Gender (male)	0.562 (0.249–1.270)	0.166		
Age	0.555 (0.376–0.820)	0.003	1.098 (0.399–3.016)	0.857
Weight	1.369 (0.944–1.986)	0.098		
Height	0.796 (0.550–1.152)	0.226		
BMI	1.627 (1.110–2.385)	0.013	0.907 (0.366–2.248)	0.832
ICMP	0.701 (0.322–1.527)	0.371		
NICMP	1.971 (0.910–4.269)	0.085		
Arterial Hypertension	0.739 (0.347–1.573)	0.433		
Diabetes mellitus	0.648 (0.302–1.391)	0.265		
Dyslipidemia	0.588 (0.270–1.282)	0.182		
Cardiovascular Disease (all)	0.442 (0.209–0.932)	0.032	0.462 (0.067–3.175)	0.432
CVD—1 vessel	1.248 (0.516–3.020)	0.624		
CVD—2 vessels	0.427 (0.115–1.587)	0.204		
CVD—3 vessels	0.393 (0.124–1.245)	0.112		
Recent MI	0.408 (0.175–0.955)	0.039	0.091 (0.012–0.667)	0.018
Recent CABG	1.039 (0.332–3.254)	0.947		
AF	0.169 (0.061–0.469)	0.001	0.028 (0.002–0.314)	0.004
COPD	0.844 (0.277–2.570)	0.766		
Asthma	4.293 (0.378–48.706)	0.240		
PAOD	1.201 (0.332–4.348)	0.780		
Anemia	0.606 (0.055–4.669)	0.548		
CKD > II	0.247 (0.109–0.563)	0.001	0.403 (0.052–3.160)	0.387
Recent Stroke	0.481 (0.128–1.804)	0.278		
NYHA (preoperative)	1.116 (0.567–2.197	0.751		
ACEI/ARB	1.001 (0.460–2.177)	0.998		
BB	0.965 (0.170–5.484)	0.968		
Ivabradine	2.796 (0.711–10.996)	0.141		
MRA	1.201 (0.527–2.735)	0.663		
ARNI	0.938 (0.418–2.104)	0.877		
SGLT2I	1.728 (0.597–5.005)	0.313		
Loop Diuretics	0.443 (0.202–0.975)	0.043	0.230 (0.040–1.319)	0.099
Digoxin/Digitoxin	0.438 (0.118–1.629)	0.218		
Amiodarone	0.144 (0.047–0.438)	0.001	0.177 (0.019–1.695)	0.133
Creatinine (baseline)	0.313 (0.164–0597)	<0.001	0.315 (0.075–1.328)	0.116
proBNP (baseline)	0.492 (0.256–0.946)	0.034	0.424 (0.038–4.686)	0.484
LBBB	2.078 (0.554–7.791)	0.278		
QRS width (preoperative)	0.928 (0.641–1.342)	0.691		
LVEF (preoperative)	0.656 (0.442–0.973)	0.036	0.497 (0.194–1.276)	0.146
LVEDD (preoperative)	0.989 (0.665–1.471)	0.957		
TAPSE (preoperative)	1.772 (1.021–3.075)	0.042	1.088 (0.399–2.969)	0.869
sPAP (preoperative)	0.999 (0.577–1.730)	0.998		
TAPSE/sPAP (preoperative)	1.626 (0.725–3.646)	0.238		
Primary Prevention	1.541 (0.521–4.559)	0.435		

CRT-D: cardiac resynchronization therapy; BMI: body mass index; ICMP: ischemic cardiomyopathy; NICMP: non-ischemic cardiomyopathy; CVD: cardiovascular disease; MI: myocardial infarction; CABG: coronary artery bypass graft; AF: atrial fibrillation; COPD: chronic obstructive pulmonary disease; PAOD: peripheral artery occlusive disease; CKD: chronic kidney disease; NYHA: New York Heart Association; ACEI/ARB: angiotensin converting enzyme inhibitor/angiotensin-II-receptor blocker; BB: beta blocker; MRA: mineralocorticoid-receptor antagonist; ARNI: angiotensin-receptor–neprilysin inhibitor; SGLT2I: sodium–glucose-transporter-2 inhibitor; LBBB: left bundle branch block; LVEF: left ventricular ejection fraction; LVEDD: left ventricular end diastolic diameter; TAPSE: tricuspid annular plane systolic excursion; sPAP: systolic pulmonary artery pressure.

**Table 8 jcm-13-04938-t008:** Univariate and multivariable binary logistic regression with regard to CRT-D responder criterion proBNP decrease ≥ 25% and various clinical characteristics.

CRT-D Responder: proBNP Binary Logistic Regression	Univariate		Multivariable	
	Hazard Ratio (95% CI)	*p*-Value	Hazard Ratio (95% CI)	*p*-Value
Gender (male)	0.356 (0.157–0.806)	0.013	0.637 (0.221–1.832)	0.403
Age	0.481 (0.322–0.717)	<0.001	0.677 (0.432–1.061)	0.089
Weight	1.353 (0.947–1.931)	0.096		
Height	0.797 (0.561–1.132)	0.204		
BMI	1.596 (1.103–2.309)	0.013	1.545 (1.023–2.332)	0.039
ICMP	0.580 (0.279–1.205)	0.145		
NICMP	1.493 (0.738–3.020)	0.265		
Arterial Hypertension	1.628 (1.628–3.389)	0.192		
Diabetes mellitus	1.253 (0.621–2.527)	0.529		
Dyslipidemia	0.920 (0.434–1.949)	0.827		
Cardiovascular Disease (all)	0.615 (0.308–1.228)	0.168		
CVD—1 vessel	2.030 (0.857–4.805)	0.107		
CVD—2 vessels	0.239 (0.065–0.884)	0.032	0.379 (0.083–1.729)	0.210
CVD—3 vessels	0.503 (0.190–1.330)	0.166		
Recent MI	0.598 (0.284–1.257)	0.175		
Recent CABG	0.583 (0.188–1.812)	0.351		
AF	0.278 (0.125–0.619)	0.002	0.369 (0.149–0.918)	0.032
COPD	0.638 (0.221–1.842)	0.406		
Asthma	2.526 (0.223–28.567)	0.454		
PAOD	0.686 (0.191–2.465)	0.563		
Anemia	0.819 (0.132–5.068)	0.830		
CKD > II	0.579 (0.287–1.164)	0.125		
Recent Stroke	0.583 (0.188–1.812)	0.351		
NYHA (preoperative)	1.317 (0.693–2.501)	0.401		
ACEI/ARB	0.445 (0.212–0.934)	0.062		
BB	1.652 (0.292–9.350)	0.570		
Ivabradine	0.598 (0.143–2.501)	0.481		
MRA	2.414 (1.072–5.435)	0.033	1.860 (0.674–5.134)	0.231
ARNI	2.890 (1.324–6.308)	0.008	2.717 (1.110–6.649)	0.029
SGLT2I	3.117 (1.017–9.551)	0.047	1.373 (0.357–5.284)	0.645
Loop Diuretics	0.268 (0.119–0.599)	0.001	0.509 (0.200–1.299)	0.158
Digoxin/Digitoxin	0.522 (0.171–1.597)	0.254		
Amiodarone	0.328 (0.147–0.734)	0.007	0.497 (0.188–1.319)	0.161
Creatinine (baseline)	0.376 (0.220–0.641)	<0.001	0.455 (0.248–0.834)	0.011
proBNP (baseline)	0.883 (0.615–1.266)	0.498		
LBBB	1.714 (0.552–5.325)	0.351		
QRS width (preoperative)	1.116 (0.789–1.577)	0.536		
LVEF (preoperative)	0.886 (0.626–1.254)	0.494		
LVEDD (preoperative)	1.045 (0.722–1.513)	0.815		
TAPSE (preoperative)	1.245 (0.772–2.008)	0.368		
sPAP (preoperative)	0.659 (0.383–1.134)	0.132		
TAPSE/sPAP (preoperative)	1.452 (0.752–2.806)	0.267		
Primary Prevention	0.621 (0.231–1.674)	0.347		

CRT-D: cardiac resynchronization therapy; BMI: body mass index; ICMP: ischemic cardiomyopathy; NICMP: non-ischemic cardiomyopathy; CVD: cardiovascular disease; MI: myocardial infarction; CABG: coronary artery bypass graft; AF: atrial fibrillation; COPD: chronic obstructive pulmonary disease; PAOD: peripheral artery occlusive disease; CKD: chronic kidney disease; NYHA: New York Heart Association; ACEI/ARB: angiotensin converting enzyme inhibitor/angiotensin-II-receptor blocker; BB: beta blocker; MRA: mineralocorticoid-receptor antagonist; ARNI: angiotensin-receptor–neprilysin inhibitor; SGLT2I: sodium–glucose-transporter-2 inhibitor; proBNP: prohormone of brain natriuretic peptide; LBBB: left bundle branch block; LVEF: left ventricular ejection fraction; LVEDD: left ventricular end diastolic diameter; TAPSE: tricuspid annular plane systolic excursion; sPAP: systolic pulmonary artery pressure.

## Data Availability

The data underlying this article will be shared on reasonable request to the corresponding author.

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
