# Peer review of "Harmonizing Heartbeats: The Mosaic of Cardiac Resynchronization Therapy Responders—A Comprehensive Exploration of Diverse Criteria and Predictors"

_jcm, 2024, doi:10.3390/jcm13164938_

Round 1

Reviewer 1 Report

Comments and Suggestions for Authors

General Comments:

This manuscript seeks to identify patient characteristics and factors associated with improved function following CRT-D implantation. Data from 132 patients implanted with CRT-D devices were retrospectively analyzed to identify associations of improved patient characteristics such as lower NYHA stage, improved LVEF, or lower proBNP decreases with pre-implant characteristics such as age, recent MI, and echocardiographic characteristics. The primary findings of the work appear to indicate that less sick patients tend to do better post CRT-D implantation. The presence of pre-implantation comorbidities such as worse RV function, older age, atrial fibrillation, and renal disease are associated with less improvement than patience without these complicating factors. This finding is not particularly novel nor surprising.

What are the takeaway messages from the paper? Do the authors propose that sicker patients or patients with certain co-morbidities should not be implanted with CRT-D devices? This paper does not identify which patients benefit most from ICD-D implantation, only that patients with identified risk factors do not do as well as patients without those risk factors. In order to identify patients that benefit from ICD-D therapy, they would need to demonstrate in matched cohorts of patients with the same risk factors have different outcomes if they receive ICD-D therapy. Therefore, neither the clinical utility nor research impact of this work appears to contribute significantly to the field.

Specific comments:

The title of the manuscript is not particularly helpful or accurate. What is being claimed by the title? What does harmonizing heartbeats have to do with this research? What is the mosaic from the title? This paper does not develop a predictive model, only shows associations.

In Table 2, what percentage of responders are categorized as responders in all 4 of these definitions? Are the same patients responders in all categories or do different patients respond with EF, proBNP, and NYHA class improvement? The implication of Table 3 is that there is probably overlap between the response characteristics, but it is not clear as to what makes the difference in response in some categories.

Figure 1 shows that responders do better than non-responders over 3 years. Since responder rates are tied to comorbidities, it seems that comorbidities lead to worse outcomes. A finding different would be surprising.

19-36: In the abstract, punctuation errors and awkward wording distract from the message. Semicolons are used in place of commas.

129-135: The list of functional status categories is confusing. Bullet points are incorrect and it is not clear why two overlapping criteria (LVEF increase of 5% and LVEF increase of 10%) are used.

In Table 2, The alignment of labels and values is very difficult to follow. The label appears to show Non-Responders first and then Responders, but then the discussion all seems to show the data the other way around. This makes interpretation of the data impossible in its current format.

While many different statistical tests are conducted, the sheer volume of the results shown makes interpretation of the results challenging. Why are 8 tables with dozens of rows and many columns needed? Increased justification for the various statistical tests and discussion of the findings are needed. For example, Table 8 results are never discussed.

Table 4 and the text show that LVEF improvement >5% and proBNP reduction were independent responder criteria, however, the interaction of these variables is not significant. Discussion of these findings would be interesting. Why is >5% LVEF significant but >10% is nowhere near significant? What does this mean? Surely these are not independent outcomes?

With the many different tests that are run, were there any corrections for multiple tests used? A p-value <0.05 is used for significance, but given hundreds of parameters tested, that could mean that 10 or more of the findings could be incorrect with a 95% confidence interval.

The discussion of the findings is broad and overreaching. What are the specific findings and measures that are of clinical significance? How can these findings impact clinical practice?

Drug use and influence is difficult to understand. The manuscript states that optical medication use is predictive of responder outcome, but use of amiodarone is associated with worse outcomes. Were drug use tracked over the 3 year followup? Did drug use change after CRT-D implantation? Were patients on more drugs at implantation simply sicker than patients with less drug use?

Creatine level was associated with responder rate, but CKD was not. Discussion of this would be beneficial.

Reviewer 2 Report

Comments and Suggestions for Authors

Boxhammer et al. in the paper entitled ,,Harmonizing Heartbeats: The Mosaic of Cardiac Resynchronization Therapy Responders — A Comprehensive Exploration of Diverse Criteria and Predictors'', proposed an ambitious, even holistic, approach to assessing response to CRT although the number of patients is not very large as well specified in the limitations of the study.

The study is well conducted but the presence of certain elements would make it much more attractive to medical professionals.

Major concern:

A. In the Introduction section, please specify why you chose to analyze only CRT-D patients and not CRT-P patients, as the end-point of interest could be common to both categories.

B. In the "Material & Methods" section

1. nothing is mentioned about the study design. A graphic abstract with the design of the study would make it much more attractive.

2. Inclusion and exclusion criteria from the study are missing.

3. Nothing is discussed about the analyzed parameters, it is not specified where they were taken from.

4. Explain what were the indications for the CRT-D implant in the case of these patients and why was CRT-P not also implanted? This explanation must be given especially since the majority of patients included in the study had NICMP (59.1%).

C. In the "Results" section, please analyze the impact on the CRT response of the following parameters.

1. CRT programming mode (biventricular pacing, LV only pacing?)

2. the impact of the malignant arrhythmic burden (if it existed!, especially since all patients were carriers of CRT-D).

Minor concerns

1. Were patients evaluated prior to implant by cardiac MRI? If so, what was the impact of myocardial fibrosis on the response to CRT?

2. Were the patients genetically evaluated especially since two thirds had NICMP? If so, what was the impact of pathogenic genes on response to CRT?

Reviewer 3 Report

Comments and Suggestions for Authors

The manuscript presents a comprehensive exploration of diverse criteria and predictors for defining responders to cardiac CRT.

Some points require further improvements to enhance the manuscript's quality. Here my concerns: How many clinicians reviewed the echocardiogram images? Did they agree on the final evaluation?

How did you calculate LVEF? The method of echocardiographic measurement should be added to the methods section.

Please explain “absence of recent MI.” What do you mean by “recent”?

The baseline table includes values prior to CRT implantation. How many patients further optimized medical therapy? This could represent an important bias.

What was the percentage of BiV or LV pacing?

Tables should be clearer (see R,NR,P)

A previous study showed that contrast-induced nephropathy after CRT could impair the recovery of EF and survival among responders. £Contrast-induced nephropathy after cardiac resynchronization therapy implant impairs the recovery of ejection fraction in responders” Please discuss this.

Previous studies have shown the impact of ventricular leads, including leadless pacemakers, on right ventricular function. “Low prevalence of new-onset severe tricuspid regurgitation following leadless pacemaker implantation in a large series of consecutive patients” “Changes in parameters of right ventricular function with cardiac resynchronization therapy”

Please include these in the discussion.

Comments on the Quality of English Language

minor english editing required

Round 2

Reviewer 2 Report

Comments and Suggestions for Authors

The changes to the article were made in accordance with the recommendations and meet the standards of scientific quality.

Reviewer 3 Report

Comments and Suggestions for Authors

congratulations to the authors for having answered to all of my comments.